# Occurrence, Genetic Variability of Tomato Yellow Ring Orthotospovirus Population and the Development of Reverse Transcription Loop-Mediated Isothermal Amplification Assay for Its Rapid Detection

**DOI:** 10.3390/v14071405

**Published:** 2022-06-27

**Authors:** Aleksandra Zarzyńska-Nowak, Daria Budzyńska, Agnieszka Taberska, Norbert Jędrzejczak, Julia Minicka, Natasza Borodynko-Filas, Beata Hasiów-Jaroszewska

**Affiliations:** Institute of Plant Protection-National Research Institute, ul. Wł. Węgorka 20, 60-318 Poznań, Poland; d.budzynska@iorpib.poznan.pl (D.B.); a.taberska@iorpib.poznan.pl (A.T.); jedrzejczak.norbert@gmail.com (N.J.); j.minicka@iorpib.poznan.pl (J.M.); n.borodynko@iorpib.poznan.pl (N.B.-F.)

**Keywords:** tomato viruses, diagnostics, TYRV, evolutionary dynamics, RT-LAMP

## Abstract

Tomato-infecting viruses have been considered as a serious threat to tomato crops in Poland. Therefore, during 2014–2021, 234 tomato samples delivered directly by greenhouse tomato growers to Plant Disease Clinic of IPP-NRI were tested. Eight virus species: pepino mosaic virus (PepMV), tomato yellow ring orthotospovirus (TYRV), tomato spotted wilt orthotospovirus (TSWV), potato virus Y (PVY), cucumber mosaic virus (CMV), tomato black ring virus (TBRV) and tomato mosaic virus (ToMV) were detected in single or mixed infection in 89 samples. The presence of TYRV was established for the first time in Poland in 2014. Since then, its presence has been observed in single and mixed infection with TSWV and CMV. Here, we analysed the genetic variability of TYRV population based on complete nucleocapsid (N) protein gene sequence of 55 TYRV isolates. Maximum-likelihood reconstruction revealed the presence of three distinct, well-supported phylogroups. Moreover, the effect of host species on virus diversity was confirmed. Therefore, RT-LAMP assay was developed for the rapid and efficient detection of TYRV isolates that can be implemented in field and greenhouse conditions.

## 1. Introduction

Tomatoes are one of the world’s most consumed vegetable crop. On a global scale, the annual production of tomatoes is estimated to be over 180 million tons and their harvesting area is still increasing [1]. After Italy, Spain, Romania, Portugal and Greece, Poland is a significant producer and exporter of tomato in UE [2,3]. The tomato yield quantity and quality depend on many factors from which diseases caused by bacteria, fungi and viruses are one of the major limiting factors influencing the production. Among the tomato pathogens, viruses are particularly difficult to control due to their high level of genetic variability, rapid evolution and adaptation as a consequence of high mutation rates, large population sizes and very short generation times that can favour the evolution of new viral variants [4]. Furthermore, plants are frequently infected by more than one virus (mixed infection), often leading to synergism, resulting in the development of very severe symptoms on infected plants and their fruits. Therefore, the detailed characterisation of viral populations provides relevant information on the processes involved in virus evolution and epidemiology which is crucial for designing reliable diagnostic tools and developing efficient and durable disease control strategies [5]. During the surveys of tomato crops conducted in the past by the Department of Virology and Bacteriology IPP-NRI in Poznań the occurrence of nine virus species: cucumber mosaic virus (CMV, genus *Cucumovirus*), pepino mosaic virus (PepMV, genus *Potexvirus*), potato virus M (PVM, genus *Carlavirus*), potato virus Y (PVY, genus *Potyvirus*), tobacco mosaic virus (TMV, genus *Tobamovirus*), tomato black ring virus (TBRV, genus *Nepovirus*), tomato mosaic virus (ToMV, genus *Tobamovirus*), tomato torrado virus (ToTV, genus *Torradovirus*) and tomato spotted wilt orthotospovirus (TSWV, genus *Orthotospovirus*) was confirmed [6,7,8,9,10,11,12]. The seasonal and spatial variation in the prevalence of viral diseases was observed, but the most frequently found viruses were PepMV and PVY in greenhouse and field tomatoes, respectively [13,14]. The reported viruses occurred in single and mixed infection and caused variable symptoms on tomato plants from very mild to even severe necrosis leading to yield and quality losses. The severity of symptoms might also be correlated with the presence of additional subviral RNAs which can be associated with the genomes of CMV and TBRV [15,16,17,18,19,20]. These particles called satellite RNAs (satRNAs) are relatively short, non-infectious and their replication, encapsidation and spread depend on the helper virus. SatRNAs share little sequence similarity with the viral genomic RNAs and their presence might have a great impact on the virus replication, accumulation and symptoms observed on infected plants [21,22,23].

In 2014, new orthotospovirus species named tomato yellow ring orthotospovirus (TYRV), was found in greenhouse tomatoes for the first time [24] and since then its presence in Poland has been observed in a single or mixed infection. The virus was first reported in Iran as tomato yellow fruit ring virus (TYFRV) in 2002 [25] and later was detected on subsequent economic plants such as potato, soya, peanut pepper, and ornamental plants such as chrysanthemum, gazania, cineraria, anemone and alstroemeria [26,27,28,29,30]. In addition, in 2012 it was detected in tomato plants in Kenya [31]. TYRV has quasi-spherical particles measuring 80–120 nm in diameter and it is transmitted naturally in a persistent propagative manner by several thrips’ species [32,33]. Similar to other orthotospoviruses, the TYRV genome contains three single-stranded (ss) RNA segments designated as small (S), medium (M) and large (L) RNA. The L RNA encodes an RNA-dependent RNA polymerase (RdRp) in a negative-sense orientation, and it is involved in transcription and virus replication. In contrast, each of the M and S RNAs consists of two genes—one in the positive- and the other in the negative-sense orientation. The M RNA encodes a non-structural protein (NSM) involved in cell-to-cell movement and glycoprotein precursor that is post-translationally cleaved into GN and GC glycoproteins associated with thrips transmission. S RNA encodes the gene silencing suppressor (NSs) and nucleocapsid (N) protein [26]. TYRV represents a new threat to the Polish tomato crops and other cultivated plant species. Typically, infected tomato plants have fruit with severe symptoms which do not possess market value (Figure 1).

It is very likely that, due to the similarity of symptoms induced by TYRV with those induced by TSWV, its presence in Poland remained unnoticed for some time. Therefore, disease management strongly relies on a fast and accurate identification of the causal agent [10]. In recent years, considerable progress has been made in developing high specificity and low detection limit tools for virus identification. Loop-mediated Isothermal Amplification (LAMP) is a frequently used nucleotide amplification technique developed by Notomi et al. [34]. It is a rapid and cost-effective diagnostic tool that enables the isothermal amplification of the target gene, using a DNA polymerase from *Bacillus stearothermophilus* that has polymerase and reverse transcriptase activity [35]. Due to this activity, LAMP can be easily combined with the reverse transcription reaction, directly detecting target RNA without a separate RT step (RT-LAMP). The LAMP reaction uses pairs of inner and outer primers that can recognize a total of six regions in the target nucleic acid. Two extra loop primers can also be employed (Loop F and Loop B) to accelerate amplification and improve detection performance [35]. The use of six primers ensures high specificity for target amplification; moreover, the loop primers ultimately accelerate the reaction and enable carrying it out within a period of half an hour in comparison when the original LAMP method is used [36]. LAMP amplifies target DNA at isothermal conditions (usually 60–65 °C) and obviates the need for thermal cyclers and postamplification procedures for signal detection. The significant advantages of LAMP assay make it very popular and widely used by many researchers to detect plant and animal viruses [37,38,39,40,41,42].

The aim of this study was the analysis of the incidence of viral diseases in commercial greenhouses in Poland with particular emphasis on orthotospoviruses. Moreover, we analysed the genetic variability and evolutionary processes involved in TYRV diversification. Analyses were performed on 55 TYRV sequences, representing the largest data set that has been analysed for this virus to date. The contribution of the host plant in TYRV population structure was investigated using Bayesian approaches. Finally, we developed an RT-LAMP assay for the rapid and efficient detection of TYRV isolates.

## 2. Materials and Methods

### 2.1. Virus Source

During 2014–2021, 234 tomato (*Solanum lycopersicum*) samples, delivered directly by greenhouse tomato growers to Plant Disease Clinic of IPP-NRI, were tested. Collected tomatoes (var. Tomimaru Muchoo, Torero) originated from twelve different Polish voivodeships. The majority of them came from Greater Poland, Kuyavian-Pomeranian and Masovian. The tested tomato plants showed variable symptoms: chlorotic or necrotic mosaic on leaves and/or fruits, necrotic plots on fruits, bright yellow ring patterns on fruits, chlorotic rings on leaves and/or fruits, leaf malformation and plant stunting suggesting virus infection. From each sample, 100 mg of apical tomato leaves were used for total RNA isolation using the RNeasy Plant Mini Kit (Qiagen, Hilden, Germany) in accordance with the manufacturer’s instructions.

### 2.2. Viruses Infecting Tomatoes in Poland

In order to identify the virus infection, the wide range of diagnostic primers detecting CMV, PepMV, PVY, TBRV, tomato brown rugose fruit virus (ToBRFV), tomato chlorosis virus (ToCV), ToMV, ToTV, TMV, TSWV and TYRV were used (Table 1). The selection of primers depended on the observed symptoms and the available information of the cultivation process (presence of vectors, symptoms, temperature conditions, origin of seedlings). Moreover, in case of samples suspected and/or confirmed to be infected with TBRV and CMV, the presence of satellite RNAs was also checked using an appropriate primers pair (Table 1). Isolated total RNAs were used in RT-PCR performed in a 50 μL mixture containing 25 μL of DreamTaq Green PCR Master Mix (ThermoFisher, Watham, MA, USA), 1 μL of RevertAid Reverse Transcriptase (ThermoFisher, Watham, MA, USA), 2 μL of specific primers pair (Table 1) and 22 μL of sterile water. The reaction conditions were as follows: reverse transcription at 42 °C for 20 min, 94 °C for 3 min followed by 30 cycles of 94 °C for 30 s, 45–66 °C (depending on the primers used) for 30 s, 72 °C for 30 s^−1^ min (depending on the primers used) and a final elongation at 72 °C for 5 min.

In a negative control reaction, total RNA isolated from healthy tomatoes was used as a template. RT-PCR products were verified by electrophoresis in 1.5% agarose gel, purified from agarose gels (Zymoclean Gel DNA Recovery Kits, Zymo Research, Irvine, CA, USA) and sequenced by an outsourced company (Genomed S.A., Warsaw, Poland). Obtained sequences were analysed using Standard Nucleotide BLAST online tool (blastn, http://blast.ncbi.nlm.nih.gov/Blast.cgi, accessed on 30 September 2021), assembled, edited and aligned using BioEdit [52] and SnapGene^®^ software (from Insightful Science; available at snapgene.com).

### 2.3. Analysis of the TYRV Polish Population

In order to analyse the virus genetic variability, full length of nucleocapsid (N) protein gene sequences of 15 TYRV Polish isolates (previously reported and three described in this study) were compared with the other 40 complete nucleocapsid (N) protein gene sequences of TYRV isolates retrieved from the GenBank database (Appendix A). The sequence of Kenyan TYRV-Loitoktok was incomplete and therefore was not included in the analysis. The multiple sequence alignments were conducted using MUSCLE [53], as implemented in MEGA X [54]. Single-nucleotide polymorphisms (SNPs) were analysed using SnapGene Viewer software. Sequence identity matrices were displayed using BioEdit and Sequence Demarcation Tool Version 1.2 (SDTv1.2) [55]. Prior to analysis of phylogenetic relationships and selection pressure, the occurrence of recombination and location of recombination breakpoints were investigated using the Datamonkey Adaptive Evolution Server with the Genetic Algorithm for Recombination Detection (GARD) method [56]. Phylogenetic analyses were carried out using the maximum-likelihood method embedded in MEGA X and appropriate nucleotide substitution model (Tamura 3-parameter (T92) with gamma distribution (+G)). Confidence in branch points in the phylogenetic trees was assessed by the bootstrap method, with 1000 pseudorandom replicates. The visualisation of the phylogenetic tree was performed using Evolview v3 webserver [57]. The pervasive and episodic selection were estimated using different methods based on the ratio of nonsynonymous and synonymous substitutions (dN/dS). Diversifying/purifying selection was estimated for individual sites using: Fixed Effects Likelihood (FEL), Fast, Unconstrained Bayesian AppRoximation (FUBAR), Single Likelihood Ancestor Counting (SLAC) and Mixed Effects Model of Evolution (MEME) statistical methods in the Datamonkey Adaptive Evolution Server [58]. The significance value was set to *p* < 0.05 for the FEL, SLAC, and MEME. The results of FUBAR, based on a Bayesian approach, were accepted when the posterior probability was greater than 0.9.

### 2.4. Phylogenetic-Trait Association

The posterior distribution of TYRV phylogenies was obtained using the Bayesian Markov chain Monte Carlo (MCMC) approach as available in the BEAST version 2.6.6 package [59]. Analyses were performed based on 50 full-length nucleocapsid (N) protein gene sequences of TYRV isolates. The minimal number of isolates representing one host plant species was two; therefore, the sequences of TYRV-Tri1M, TYRV-Tob2M, TYRV-t/TR-Gaz8, TYRV-A/TR-Ane4 and TYRV-CI/TR-Cin5 isolates were deleted from the analyses. Firstly, the multiple sequence alignment was performed using the MUSCLE algorithm as implemented in MEGA X. Then, to find the best-fitting substitution model for our data the jModelTest was applied [60]. Based on Bayesian information criterion (BIC), Hasegawa-Kishino-Yano (HKY) model with gamma distribution (+G) was chosen as the most proper one. Subsequently, the input files for BEAST analyses were created in BEAST-associated program BEAUTi. Both strict and relaxed molecular clock models and different tree priors were tested and then the obtained results were compared to choose the most adequate combination for our data. The final analysis was performed with the relaxed clock exponential model and coalescent constant population tree prior. The chain length was run for 10^7^ generations with first 10^4^ samples discarded as a burn-in. Convergence of the particular parameters and their effective sample size (ESS > 200) was checked with Tracer v1.7.2 [61]. Finally, maximum clade credibility tree (MCC) was annotated and summarised from the posterior distribution of trees generated by BEAST (after ignoring 10% of trees) using TreeAnnotator v 2.6.3 (as a part of BEAST package). An MCC tree was visualised with FigTree v 1.4.4 (http://tree.bio.ed.ac.uk/software/figtree/, accessed on 25 November 2018). Subsequently, phylogeny-trait association was evaluated with BaTS 2.0 [62]. As a trait, the host origin of TYRV isolates was defined. The analysis was performed based on 9001 sample trees generated by BEAST (after removing first 10% of trees). To investigate the host origin distribution along the phylogenetic tree, three different statistics were obtained: parsimony score (PS), association index (AI) and the maximum monophyletic clade size (MC). Both AI and PS enabled us to investigate the overall degree of the host plant structure among the TYRV population, whereas particular MC values allowed us to assess the level of clustering in individual host plants. The values of PS, AI and MC were assigned as statistically significant when the *p*-value < 0.05.

### 2.5. RT-LAMP

Eight TYRV isolates were selected for RT-LAMP development including three described in this study. Total RNAs extracted from healthy and infected with TSWV tomato plants were used as negative controls. In the first step, the specific primer pairs were designed based on the complete genome alignment of TYRV isolates nucleocapsid (N) protein gene sequences retrieved from the GenBank database (Appendix A). The set of diagnostic primers (Table 2) was designed using LAMP Designer (OptiGene, Horsham, UK). The RT-LAMP was performed in a total volume of 25 μL. The reaction mixture contained 15 μL of Isothermal Mastermix (ISO-001nd) (Novazym, Poznań, Poland), 0.5 μL of reverse transcriptase (Roche, Basel, Switzerland), 2 μM of FIP and BIP, 0.5 μM of F3 and B3 primers, 1 μL of tested RNAs and 3.5 μM of sterile water. In order to optimize the reaction conditions, the mixture was incubated at 60 °C, 63 °C, 65 °C and 70 °C for 30 min and 1 h in Biometra T3000 thermocycler (Biometra, Göttingen, Germany). The RT-LAMP products were analysed by electrophoresis in 2% agarose gel and directly by the visual inspection of the reaction tube under UV light after adding 2 μL of EvaGreen^®^Dye (Biotium, Fremont, CA, USA). In addition, to establish primer specificity, the RT-LAMP was performed using LightCycler ^®^96 Instrument (Roche, Basel, Switzerland). The reaction consisted of 7.5 μL of Isothermal Mastermix Fluorescent Dye (ISO-001) (Novazym, Poznań, Poland), 1 μM of FIP and BIP, 0.25 μM of F3 and B3 primers, 0.25 μL reverse transcriptase (Roche, Basel, Switzerland) and 0.5 μL of target RNA in a final volume of 12.5 μL. The fluorescence signals were recorded in FAM channel (excitation at 470 nm, detection at 510 nm) for 30 min. The sensitivity of RT-LAMP assay and conventional RT-PCR were estimated and compared. For this purpose, tenfold dilutions of TYRV-H total RNA were used. The RNA concentration was measured using a NanoDrop 2000 spectrophotometer (Thermo Fisher Scientific, Waltham, MA, USA), adjusted to 100 ng/μL, and serially diluted to serve as a template for both RT-LAMP and conventional RT-PCR methods. The RT-PCR was performed using 2 μL of TY2-F/TY2-R primers (Table 1), 1 μL of diluted RNA, 25 μL of DreamTaq Green PCR Master Mix (Thermo Fisher Scientific, Waltham, MA, USA), 1 μL RevertAid Reverse Transcriptase (Thermo Fisher Scientific, Waltham, MA, USA) and 22 μL of sterile water. The reaction conditions were as described hereinabove. The detection limit of compared techniques was determined by electrophoresis in 1.5% agarose gel.

## 3. Results

### 3.1. Viruses Infecting Tomato in Poland

Out of 234 collected tomato samples, 89 were infected by viruses in single or mixed infection (Table 3). The results of RT-PCR were confirmed by sequencing. Predominantly detected virus was PepMV (66.2%), found in tomatoes every year. The results also indicated a high incidence of orthotospoviruses, followed by TYRV (16.9%) and TSWV (13.4%). CMV and PVY were detected at a lower frequency of 7.8% and 2.2%, respectively, whereas TBRV and ToMV were indicated only once. Mixed infection with two of the above- mentioned viruses were observed in 9% of the analysed samples. In case of mixed infection, the symptoms observed on the tested plants were usually more severe. Samples originated from tomato plants which exhibited necrotic mosaic on leaves and considered to be infected with CMV or TBRV were also tested for the presence of satRNAs. The presence of satRNAs was not confirmed in CMV neither TBRV-infected samples.

### 3.2. Analysis of TYRV Population

The nucleocapsid (N) protein gene sequences of three new Polish TYRV isolates (TYRV-H, TYRV-KOR1, TYRV-KOR2) of 825 nucleotides in length were deposited in GenBank under accession numbers: OM654405-OM654407, respectively. The nucleotide and amino acid sequence identity of the 15 Polish TYRV nucleocapsid (N) protein gene sequences were relatively high and ranged from 99.2% to 99.8% and from 99.2% to 100%, respectively (Appendix A). A comparative study of nucleocapsid (N) protein gene sequences of the Polish isolates (compared to TYRV-TK1 sequence) revealed the presence of several point mutations from which 15 were nonsynonymous ones (Appendix A). The level of genetic diversity within the Polish TYRV population was rather low, but the Polish isolates differ significantly from those collected in Iran. The nucleotide and amino acid nucleocapsid (N) protein gene sequence identity of all 55 TYRV isolates ranged from 83.9% to 100% and from 89.4% to 100%, respectively (Figure 2 and Appendix A). The sequences of TY-PH60, TY-PE14, TY-PH132 and TY-PK95 were identical, whereas those belonging to the isolates TYRV Tob2-M, TYRV-Cl and TYRV-Tri1M were characterised as the most divergent ones. Recombination analysis showed no evidence of recombination events within the analysed population. Phylogenetic analysis performed on the nucleotide sequences revealed defined clustering by geographical origin (Figure 3). Three geographical subpopulations can be distinguished: cluster of the Polish isolates (POL) and two separate clusters gathering Iranian isolates (IRN-1 and IRN-2). Newly identified TYRV isolates were closely related to the ones previously reported in Poland. Interestingly, no clear clustering according to the host plant was observed. Isolates originated from *S. lycopersicum* grouped in POL and IRN-1 clusters, whereas isolates from *S. tuberculosum* and *Glycine max* were located in both Iranian clusters. Nevertheless, isolates from *Arachis hypogaea, Capsicum annum, Alstroemeria* and *Chrysanthemum* isolates grouped together within appropriative Iranian clusters. All isolates from ornamental plants were found in IRN-1 cluster. Selective pressure analysis indicated that TYRV population is driven by purifying selection. The evidence of potential negative selection was obtained for 87 codons, from which 52 were detected by three of the used algorithms (FEL, SLAC, FUBAR) (Appendix A). Only three codons were identified by two algorithms to be under positive selection: codon 2, 57 and 185.

### 3.3. Host Driven Structure of the TYRV Population

In order to identify the host-driven structure in the TYRV population, firstly, the Bayesian phylogeny for 50 nucleocapsid (N) protein gene sequences was inferred (Appendix A). Secondly, three summary statistics were calculated: AS, PS and MC. The values of AS, PS and MC and their statistical support were presented in (Table 4). Analyses included TYRV isolates originated from: *S. lycopersicum* (22 isolates), *Alstroemeria* sp. (2 isolates), *Chrysanthemum* sp. (2 isolates), *G. max* (2 isolates), *S. tuberosum* (14 isolates), *A. hypoagea* (6 isolates) and *C. annuum* (2 isolates) (Suplementary Appendix A). The sample size required for analyses was at least two and, therefore, the isolates from *Tropaeolum majus* (1 sample), *N. tabacum* (1 sample), *Gazania* sp. (1 sample), *Anemone* (1 sample) and *Cineraria cruenta* (1 sample) were excluded due to an insufficient sample size. Despite the reduced number of isolates used in analyses, the topology of obtained MCC tree was consistent with this presented by a maximum likelihood tree (Appendix A). A phylogenetic trait association test allowed us to confirm that there is an overall effect of host species in the distribution of TYRV variability, as shown by significant AI and PS values (*p*-value < 0.00). Nonetheless, only two subpopulations of TYRV isolates retrieved from *S. lycopersicum* and *S. tuberosum* showed differentiation, whereas no statistically significant results were obtained for *Alstroemeria* sp., *Chrysanthemum* sp., *G. max, A. hypoagea*, and *C. annuum* (Table 4).

### 3.4. Optimisation of the RT-LAMP Conditions

RT-LAMP primers designed in this study efficiently amplified all the tested TYRV isolates (Figure 4A). The amplification products, visible as ladder-like DNA fragments, were obtained in all the tested temperatures and time periods, but the most optimal conditions were 63 °C and 60 min of reaction time. Samples from TSWV-infected and healthy plants were negative. Green colour, indicating a positive reaction, was observed in UV light after adding the dye only for the TYRV-infected samples and no colour changes were seen for the negative controls (Figure 4B). A sensitivity test for the RT-LAMP method revealed it was ten times more sensitive than conventional RT-PCR (Figure 4C,D).

## 4. Discussion

Plant viral diseases cause epidemics in crops that are often too complex to control and threaten food security. This difficulty is mainly due to the lack of early detection and ineffective countermeasures, especially for emerging viruses. Furthermore, mixed infection affecting a single plant or crop are recognised to be common in plant disease epidemics. Mixed infection can generate a range of ecological interactions that have an impact on viral fitness, and consequently have far-reaching epidemiological implications. From over 130 viruses knowing to infect tomatoes, about 50 species are considered as serious threat causing several problems in field-grown and greenhouse crops [63,64]. During 2014–2021, in order to analyse the occurrence of viruses infecting greenhouse tomato in Poland 234 plant samples were tested. The majority of the collected samples with virus-like symptoms were not infected with any of tested viruses. It is likely that visible symptoms were caused by physiological plant issues, inadequate dose of fertilisers or inappropriate use of plant protection products. We also cannot exclude the possibility that samples were infected with other virus species, designed and used primer pairs were not complementary to present viral isolates or the virus concentration in tissue was too low to be detected using conventional PCR. These problems can be solved by using high-throughput sequencing (HTS), which was proved to be very successful for virus discovery to resolve disease etiology in many agricultural crops. The greatest advantage of HTS over other diagnostic approaches is that it gives a complete view of the viral phytosanitary status of a plant [65]. As a diagnostic tool, HTS is perhaps more broad-spectrum than any previously used assay. Nevertheless, using HTS for routine virus detection is limited due to high costs, equipment requirement and computing equipment capable of storing and processing large datasets, as well as expertise on genome assembly analysis pipelines and packages [66].

Among infected samples, PepMV was the most frequently detected. It is a rapidly emerging virus, which has been established as one of the most important viral diseases in tomato production worldwide [63]. In Poland, several isolates belonging to the European and Chilean 2 genotypes have been found since 2003 [8]. Since then, the virus has been constantly detected in tomato crops in single and mixed infection [8,67]. In 2017 and 2018, plant protection products PMV^®^-01 of Belgian DCM company and V10 of Dutch Valto company were registered in Poland, respectively. The vaccination strategy is based on the principle of cross-protection, where a mild isolate of the virus is used to protect plants against more severe ones. Since then, the number of PepMV-infected samples has been decreasing in comparison to previous years. Nevertheless, PepMV still remains one of the most important threats to tomato cultivation due to its high genetic variability, quasispecies nature and creation of new variants by mutations and recombinations [8,68]. TYRV in single and mixed infection with TSWV was found in tomato plants for the first time in 2014 [24]. Orthotospoviruses are one of the major threats to vegetable crops worldwide. Research conducted in Plant Disease Clinic of IPP-NRI, Poznań, revealed that TSWV has been mostly identified in ornamental plants such as chrysanthemum or gerbera. On the other hand, in the following years, its presence was also confirmed in tomato, and the infected fruits did not have any commercial value. Apart from TSWV, TYRV was also detected in a mixed infection with other orthotospoviruses such as iris yellow spot orthotospovirus (IYSV) and impatiens necrotic spot orthotospovirus (INSV) [69]. In 2021, otherwise than earlier, TYRV was detected in a mixed infection with CMV, which is the first evidence of a TYRV infection with the virus representing a different genus. CMV (genus *Cucumovirus*) has one of the broadest host range among plant viruses and is responsible for important agronomic losses in many crops worldwide [70]. During tomato surveys in Poland, CMV has been occasionally detected, in single infection in four samples and in mixed infection with PepMV and TYRV. Co-infection with CMV and viruses from another genus (such as *Crinivirus, Potexvirus, Potyvirus, Tobamovirus*) led to synergists interactions affecting the symptoms development and increasing the CMV accumulation [70]. In Poland, CMV is definitely more often identified on cucumber and zucchini than tomato [71,72]. What is more, the CMV genome might be associated with satRNAs which are responsible for the development of a lethal disease in tomato. Epidemics of satellite-RNA-containing isolates have been reported in France, Spain, Italy and Greece [73,74,75,76]. The D-satRNA, B-satRNA, and WL1-satRNA induce necrosis, chlorosis, and attenuation, respectively [23]. Therefore, tomato samples exhibited necrotic mosaic on leaves and identified to be infected with CMV were additionally tested for the presence of satRNAs but no additional subviral particles were detected. Another virus species identified in tomato population was PVY. It was first reported in tomato greenhouse production in 2002 [10] and since then the presence of the virus in Poland has been occasionally detected. The situation changed in 2008 when the increased distribution of the PVY^N^Wi-P strain was observed [77]. The surveys conducted in greenhouse and fields during 2012–2013 revealed that PVY population is much more diverse and the isolates representing different strains were identified: PVY^NTN^, PVY^0^, PVY^N^Wi-P, PVY^N^N242 and new recombinant variant between PVY^NTN^ and PVY^N^Wi-P. The predominant strain in greenhouse population was PVY^N^Wi-P and this strain was also identified in 2020 suggesting its prevalence in Poland. The remaining viruses: TBRV and ToMV were detected only once, and it seems that they are not a threat to tomato production. Tobamoviruses on tomato have been rarely detected but lately they were identified in irrigation ditches and drainage canals surrounding fields [78] which can be potential virus source for tomato field infection. In case of TBRV, the presence of satRNAs was also verified as they might have an impact on virus replication and accumulation. None of the subviral RNAs were detected in the tested sample.

Recently, new emerging virus species have been identified in the EU. One of them was ToBRV, firstly reported in a greenhouse tomato crop in 2014 in Jordan [79]. To date, the incidence of ToBRV infection has been reported in 34 countries, including Poland, where the virus was found for the first time in tomato plant and seeds in 2020 [80]. The second one is ToCV, which is considered as a serious production problem for field and greenhouse tomato growers [81]. Therefore, tomato samples, which exhibited symptoms such as mild foliar symptoms with clearly brown rugose symptoms on fruits, interveinal chlorosis, leaf brittleness and limited necrotic flecking or leaf bronzing on leaves, were tested for the occurrence of those viruses. No samples were confirmed to be infected with ToBRFV and ToCV.

Pathogen diagnostics plays an important role in crop protection and disease management. The detailed analysis of genetic structure of virus population is essential to develop effective, sensitive, low-time and cost detection method to identify viruses especially when new pathogen species emerged. Therefore, in this work we decided to analyse the evolutionary dynamics of TYRV which presence has been observed in Poland since 2014. The phylogenetic analyses divided TYRV populations into three clusters corresponding to geographical regions of their origin. The phylogenetic division into two Iranian clusters was reported previously [30,82]. Based on the results obtained by Golnaraghi et al. the isolate from Kenya grouped in Iranian cluster 2 [30]. The Polish isolates came from two neighbouring provinces known for tomato greenhouse production (Kuyavian-Pomeranian and Greater Poland) [50], whereas Iranian ones were collected from distinct provinces such as Fars, Teheran, Markazi, Mazandaran [25,26,28,58]. These distant locations suggest virus transmission with plant propagative materials [83]. Additional analyses revealed that the TYRV population exhibits some host-driven structure, especially in reference to isolates originated from tomato and potato. These results generally correspond to grouping on the phylogenetic tree, but they should be interpreted with caution due to the overabundance of tomato and potato isolates in comparison to the representants of other host plants.

The genetic structure and diversity of viral populations can be affected by several factors, such as recombination and selective pressure. Recombination clearly plays a significant role in the evolution of RNA viruses by generating genetic variation, and is beneficial in the case of genes that are under diversifying selection [84]. The recombination analysis performed based on full-length nucleocapsid (N) protein gene sequences of 55 TYRV isolates showed no evidence of recombination. In addition, within this population, purifying selection was found to be predominant. It is assumed that purifying selection acts on codons, in which change may adversely affect the properties of the virus and, therefore, genetic variants with these mutations will be eliminated from the virus population. Similar results were obtained by Golnaraghi group [30] who used a smaller number of isolates and different methods to analyse the TYRV population structure. In the same work, the authors indicated that the TYRV Polish population recently increases in size preceded by a genetic bottleneck, i.e., a founder effect [30].

Disease management strongly relies on a fast and accurate identification of the causal agent [85]. Several diagnostic techniques have been developed for TYRV detection including both serological and molecular methods [31,51,86]. Double antibody sandwich enzyme-linked immunosorbent assay (DAS-ELISA) is widely used in large-scale screening for TYRV infection in various types of plants; however, the molecular techniques are more sensitive than the serological ones. Low virus titre can limit sensitivity, producing false negatives when the virus concentration is under the technique detection threshold [85]. Therefore, we aimed to develop an RT-LAMP assay for the reliable and rapid diagnostics of TYRV. LAMP exhibits a higher sensitivity in comparison to PCR and is less affected by inhibitors present in tomato plants (e.g., phenols and complex polysaccharides), which are often a cause of false negative results. LAMP assay does not require a highly purified nucleic acid template and it also may be performed effectively with crude extract from an infected plant. Therefore, it can be carried out using lateral flow devices, making it suitable for onsite detection or testing in the field. The products of LAMP can be detected much faster than in standard techniques, sometimes only requiring analysis with the naked eye [87]. It has been reported that LAMP assay can be even 1000 times more sensitive than conventional PCR with equivalent specificity [88]. The primers for RT-LAMP were designed based on the alignment of the complete nucleocapsid (N) protein gene of TYRV in order to amplify the wide range of TYRV isolates. RT-LAMP assay developed in this study is capable of detecting TYRV isolates within 1 h. The result of RT-LAMP was direct visualized under UV light by adding fluorescent dye and in real-time conditions. The sensitivity was tenfold higher than that of RT-PCR. Our results indicate that the TYRV RT-LAMP assay is sensitive and specific, and has the potential to be developed into a field or greenhouse diagnostic test.

## Figures and Tables

**Figure 1 viruses-14-01405-f001:**
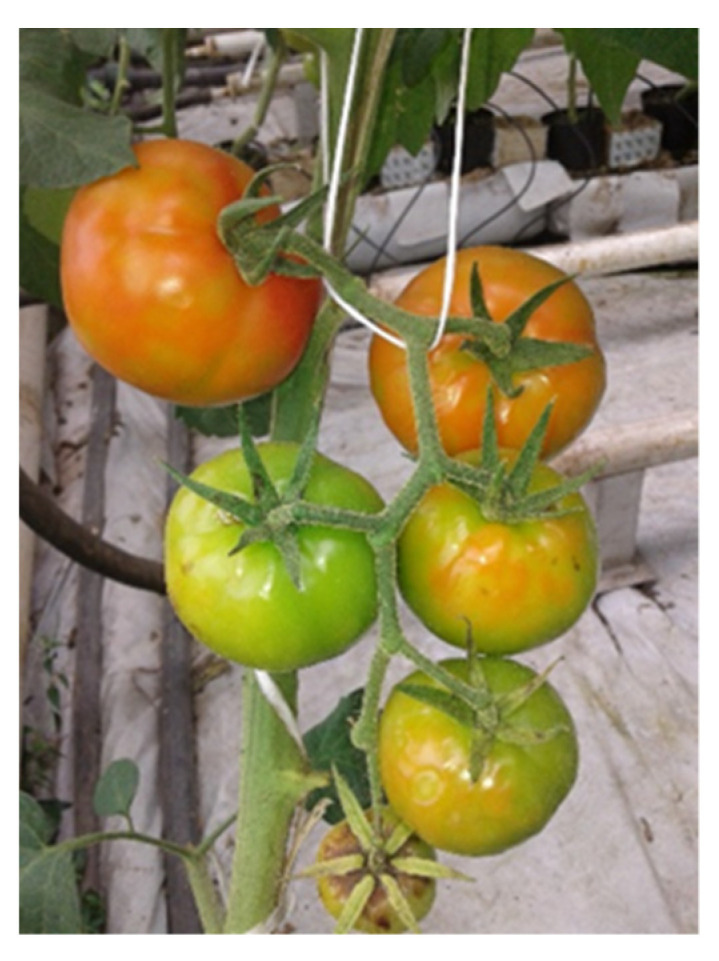
Tomato fruits grown in commercial greenhouse infected with TYRV.

**Figure 2 viruses-14-01405-f002:**
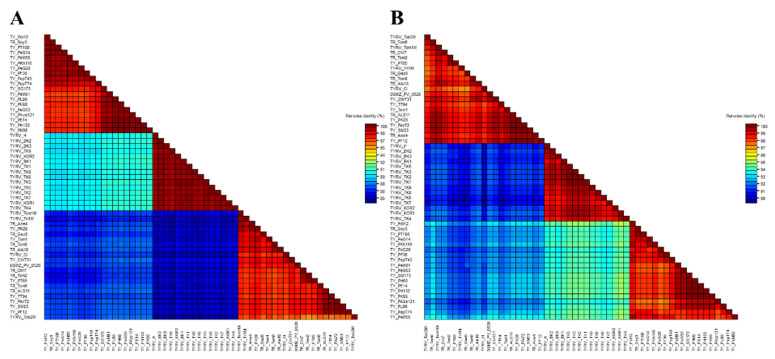
Two-dimensional visualization of nucleotide (**A**) and amino acid (**B**) nucleocapsid (N) protein gene sequence identity of 55 tomato yellow ring virus (TYRV) isolates used in this study. The matrices were performed using SDTv1.2.

**Figure 3 viruses-14-01405-f003:**
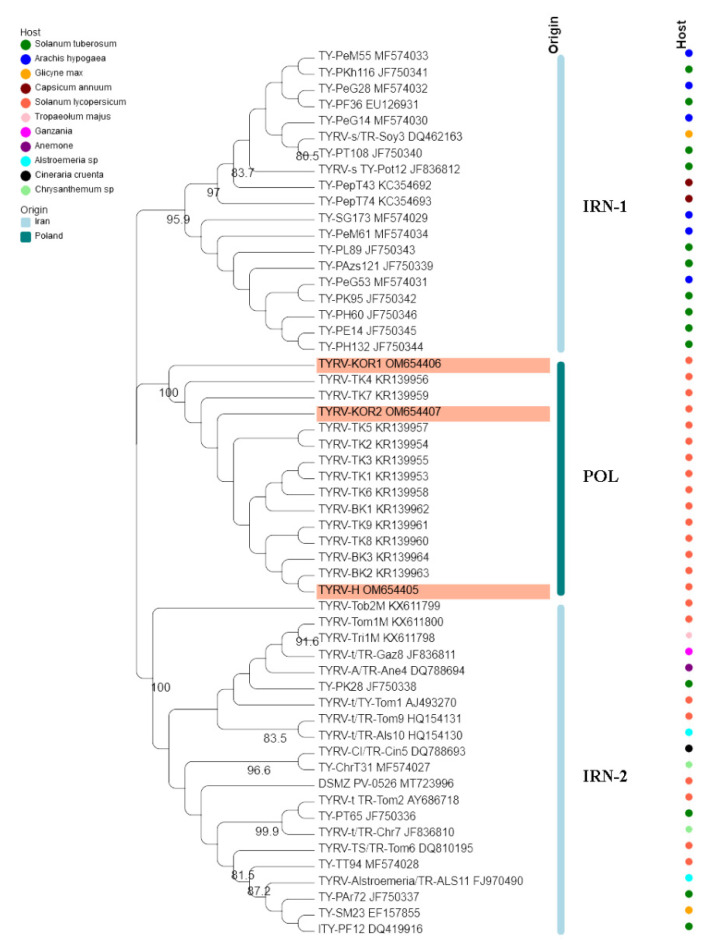
Phylogenetic tree constructed based on nucleocapsid (N) protein gene nucleotide sequences of 55 tomato yellow ring virus (TBRV) isolates. Double names of some isolates correspond to names of isolates in GenBank and those used by Golnaraghi et al. in their work [30]. Host species and country of origin were represented by dots and lines, respectively. Newly described Polish TYRV isolates are highlighted.

**Figure 4 viruses-14-01405-f004:**
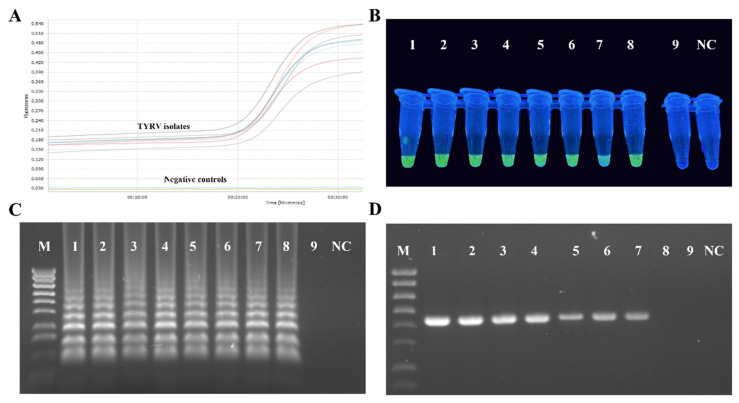
Results of RT-LAMP optimalization. (**A**) Amplification plots (showed with different colours) of eight tested TYRV samples monitored in the LightCycler^®^ 96 (Roche, Basel, Switzerland). Total RNAs isolated from healthy and TSWV-infected plants were used as negative controls. (**B**) Visual detection of RT-LAMP products using EvaGreen^®^Dye (Biotium, Fremont, CA, USA) of eight TYRV isolates (TYRV-TK1, TYRV-TK2, TYRV-TK4, TYRV-BK1, TYRV-BK2, TYRV-H, TYRV-KOR1, TYRV-KOR2, probes 1–8, respectively), TSWV isolate (probe 9) and healthy plant (NC). (**C**) Electrophoretic separation of RT-LAMP products in a 2% agarose gel. M- Gene Ruler 1 Kb Plus Ladder (ThermoFisher Scientific, Watham, MA, USA), lines 1–9—ten-fold dilutions of total RNA of TYRV-H isolates, NC-negative control. (**D**) Electrophoretic separation of RT-PCR products in 1.5% agarose gel. M-Gene Ruler 1 Kb Plus Ladder (ThermoFisher Scientific, Watham, MA, USA), lines 1–9 ten-fold dilutions of total RNA of TYRV-H isolates, NC-negative control.

**Table 1 viruses-14-01405-t001:** Primers used in tomato virus diagnostics.

Virus Name		Primer Name	Primer Sequence 5′-3′	Annealing Temp.	Reference
cucumber mosaic virus	CMV	CMVCPf	GCTTCTCCGCGAG	50 °C	[43]
		CMVCPr	GCCGTAAGCTGGATGGAC		
cucumber mosaic virus		CMVsat-fwd	AAGGATCCGGGTCCTGBDDDGGAATG	55 °C	[44]
satRNA		CMVsat-rev	AAGGATCCGTTTTGTTTGWTRGAGAAT TGCGYRGAG		
pepino mosaic virus	PepMV	TGB3F	GGTGGACAATATCAAGACGG	51 °C	[41]
		TGB3R	CTGTATTGGGATTTGAGAAGTC		
potato virus Y	PVY	PVYc3F	CAACGCAAAAACACTCAyAAAmGC	57 °C	[45]
		PVYfR	TAAGTGrACAGACCCTCTyTTCTC		
		PVY3F	TGTAACGAAAGGGACTAGTGCAAAG		
		PVY3R	CCGCTATGAGTAAGTCCTGCACA		
		PVYCP2F	CCAGTCAAACCCGAACAAAGG		
		PVYCP1R	GGCATAGCGTGCTAAACCCA		
tomato black ring virus	TBRV	TBRVR1-P1-KRF	GGTAAAAGTTCTGGGTGCT	53 °C	[46]
		TBRVR1-P1-KRR	GCAAATCCACCTCCTTATCC		
tomato black ring virus		CH_SAT_F1	TAATTTTGAAAGTCTCTGA	47 °C	[46]
satRNA		CH_SAT_R2	GGACAGCTCGTTGGTTCTTAGA		
tomato brown rugose	ToBRFV	CaTa28 Fw	GGTGGTGTCAGTGTCTGTTT	60 °C	[47]
fruit virus		CaTa28 Rv	GCGTCCTTGGTAGTGATGTT		
		CaTa28 Pr	6FAM-AGAGAATGGAGAGAGCGGACGAGG-BHQ’1		
		CSP1325 Fw	CATTTGAAAGTGCATCCGGTT T		
		CSP1325 Rv	GTACCACGTGTGTTTGCAGACA		
		CSP1325 Pr	VIC-ATGGTCCTCTGCACCTGCATCTTGAGA-BHQ’1		
tomato chlorosis virus	ToCV	ToCVCPF	ATGGAGAACAGTGCCGTTGC	58 °C	[48]
		ToCVCPR	TTAGCAACCAGTTATCGATGC		
tomato mosaic virus	ToMV	ToMV F	CGAGAGGGGCAACAAACAT	66 °C	[49]
		ToMV R	ACCTGTCTCCATCTCTTTGG		
tomato torrado virus	ToTV	2TT5	GATGAGAAAGGAAAGAAGCAG	55 °C	[50]
		2TT6	CATATCACCCAAATGCTTCTC		
tobacco mosaic virus	TMV	TMV F	CGACATCAGCCGATGCAGC	66 °C	[49]
		TMV R	ACCGTTTTCGAACCGAGACT		
tomato spotted wilt	TSWV	TS1-F	GCCTATGGATTACCTCTTG	45 °C	[51]
orthotospovirus		TS1-R	GTTTCACTGTAATGTTCCA		
tomato yellow ring	TYRV	TY2-F	CTAACAAAGCCATGAAGA	45 °C	[51]
orthotospovirus		TY2-R	GAAGACCCAGCACCA		

**Table 2 viruses-14-01405-t002:** Primers used in RT-LAMP reaction.

Primers Name	Primer Sequence 5′-3′
TYRV FIP	CACAGTAGAGCTAGGAACAACAATA-AAAATGGTTAAAGCAGGGC
TYRV BIP	GGTCAAGATGATTGGACATTCCGA-TGCATTTTCCACAGCAATG
TYRV F3	GAGAAACAGAGCAGGGATT
TYRV B3	TCATACATTTTCTGTTTCTCAGT

**Table 3 viruses-14-01405-t003:** Tomato samples tested by RT-PCR for 11 virus species grouped by year.

Year	No. of Collected Samples	No. of Infected Samples	Single Infection											Mixed Infection			
			PepMV	TYRV	TSWV	CMV	PVY	ToMV	TBRV	TMV	ToTV	ToBRV *	ToCV *	TSWV+ TYRV	CMV+ PepMV	TSWV+ PepMV	CMV+ TYRV
2014	54	25	7	8	6	-	-	-	-	-	-	x	x	4	-	-	-
2015	41	11	11	-	-	-	-	-	-	-	-	x	x	-	-	-	-
2016	37	11	10	-	-	1	-	-	-	-	-	x	x	-	-	-	-
2017	22	6	5	1	-	-	-	-	-	-	-	x	x	-	-	-	-
2018	24	6	6	-	-	-	-	-	-	-	-	x	x	-	-	-	-
2019	20	5	3	-	-	2	-	-	-	-	-		x	-	-	-	-
2020	18	11	7	-	-	-	2	-	-	-	-	-	-	-	2	-	-
2021	18	14	7	1	1	1	-	1	1	-	-	-	-	-	-	1	1
Total	234	89	56	10	7	4	2	1	1	0	0	0	0	4	2	1	1

* The virus presence has been tested since its first report in neighbouring countries in 2018 and 2020.

**Table 4 viruses-14-01405-t004:** Bayesian statistic measuring the association between host plant and clustering observed in the MCC tree. Statistical supported values are bolded.

Isolate Origin (Host Plant)	Observed Value			Null Value			*p*-Value
	Mean	Lower HPD	Upper HPD	Mean	Lower HPD	Upper HPD	
		95% CI	95% Cl		95% CI	95% CI	
**AI**	**2.96**	**2.45**	**3.47**	**4.00**	**3.49**	**4.59**	**0.00**
**PS**	**18.21**	**17.00**	**19.00**	**24.93**	**22.74**	**26.82**	**0.00**
**MC (SL)**	**15.00**	**15.00**	**15.00**	**2.70**	**2.00**	**3.66**	**0.01**
MC (AL)	1.00	1.00	1.00	1.01	1.00	1.09	1.00
MC (CH)	1.00	1.00	1.00	1.01	1.00	1.06	1.00
MC (GL)	1.00	1.00	1.00	1.01	1.00	1.07	1.00
**MC (ST)**	**3.11**	**2.00**	**5.00**	**1.89**	**1.28**	**2.52**	**0.04**
MC (AR)	1.08	1.00	2.00	1.21	1.00	2.00	1.00
MC (CA)	1.00	1.00	1.00	1.02	1.00	1.04	1.00

## Data Availability

Not applicable.

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
