# Peer review of "Occurrence, Genetic Variability of Tomato Yellow Ring Orthotospovirus Population and the Development of Reverse Transcription Loop-Mediated Isothermal Amplification Assay for Its Rapid Detection"

_viruses, 2022, doi:10.3390/v14071405_

Round 1

Reviewer 1 Report

Dear Authors,

In your article you surveyed tomato plants for the presence of tomato infecting viruses by traditional RT-PCR method. Investigating more than 200 plants in the last 8 years you found infection with 7 viruses alone or in combinations. You find quite a high incidence of infection with tomato yellow ring orthotospovirus (TYRV), a relatively newly described virus. Phylogenetic analysis of TYRV revealed that variants cluster according to geographical origin rather than according to the hosts. RT-LAMP primers were designed and tested to be successfully detect the virus. Moreover, comparison of RT-LAMP with RT-PCR showed that the previous methods is more sensitive.

I liked your manuscript. I think that the Introduction and the discussion part (about the virus diagnostics) is very nicely and clearly written. The results are convincing. Based on the title I anticipated a much longer discussion and methodology about the RT-LAMP design and optimization, I think this is weakest part of the manuscript.

Your manuscript is interesting and, in my opinion, it is suitable for publication in MDPI Viruses. I would suggest you some editing and correction points, which I think would further increase the quality of the paper when it will be published.

These points and critics are:

1/ Abstract

The first half of the abstract is like an introduction. I would suggest to rewrite it focusing on the new results and keep only the absolutely essential knowledge here, about the survey you have done (how long, how many plants were tested and how many viruses were found). Phylogenetic clustering was according to the geographical origin, evolution seems to favour purifying selection and what part of the genome was used for RT-LAMP primer design, which was tested on how many isolates and was compared to RT-PCR.

2/ Introduction

line 50: I would consider to cite https://doi.org/10.1371/journal.ppat.1002021 describing molecular mechanism behind symptom development in Y sat containing CMV infection.

Line 57: emphasize that TYRV has only been described from Poland, Iran and Kenya. It is not clear here and later on it was not straightforward to understand why you explain why the isolate from Kenya was not included in the analysis.

3/ Materials and methods

2.2 Was RT-PCR done as a one step reaction? If you tested one sample for the presence of several viruses it would have been more valid to use the same cDNA for all of the virus test.

line 144: instead of “deleted from the set” I would suggest “was not included in the analysis”

Table2: I would suggest to put it into the Supplementary materials

2.5 RT-LAMP: it seems that only one sets of LAMP primers were designed by the software.

It is not too much description on this primer design. Even if the software would have been designed the primers it could be aligned (not only to the Polish, but maybe for the Iranian and Kenya variants) to show how conservative the stretches of the sequence are what accommodate the primers. You made temperature gradient, but increased the temperature by quite high (3°C) steps. Wouldn’t it be better to slightly increase the temperature? keeping in mind that it is already a high temperature for disrupting the secondary structured RNAs? It is not stated how many times were the LAMP test carried out? This would be important to see the reproducibility of the assay. No characteristic parameters: sensitivity, reliability, reproducibility was investigated, this would be nice to do. You only used RNA for the LAMP test. As it can operate on crude extract it would further simplify its use. If you did not test that, please at least discuss this possibility.

4/ Results

Table 4: in this table I would change the order of the viruses and list them according to their incidence, as it was done in the discussion. This would make this table and the discussion parallel, and help to understand the numbers. I would include the satellites also, just after their helper virus.

Figure 3 and results of the phylogenetic analysis. Your statement is that the strains cluster according to their geographical origin. The trees clearly show this, but I am curious: to which ones the isolate from Kenya would cluster. It would be possible to make a tree including 2-3 isolates from both 3 clusters (POL, IRN-1 and IRN-2) and cut them to the same size what the Kenya isolate has and make a tree.

Comparison of RT-LAMP with RT_PCR. It is not clear from the shown pictures how the 10x increased sensitivity was determined. Where is nu9 on Fig 4/D?

5/ Discussion

Negative result for the investigated viruses with RT-PCR can be the result of SNP variants which has mutation withing the primers sequence. This is something what you should at least discuss.

Is it possible to check back the seeds for the presence of the virus, to find out its origin? It is said in the M&M that the tomato cultivars were mainly Polish origin, while in the discussion the seeds imported from Israel and China are suspected as the source of the virus. This should be clarified.

I think that after clarifying these things and making some correction the manuscript will be acceptable for publication.

Author Response

Reviewer 1

We would like to thank you for all valuable comments. We revised the paper taking into account all your suggestions. Please, find below detailed response for all questions.

  1. The first half of the abstract is like an introduction. I would suggest to rewrite it focusing on the new results and keep only the absolutely essential knowledge here, about the survey you have done (how long, how many plants were tested and how many viruses were found).

Authors: The abstract was changed.

  1. Phylogenetic clustering was according to the geographical origin, evolution seems to favour purifying selection and what part of the genome was used for RT-LAMP primer design, which was tested on how many isolates and was compared to RT-PCR.

Authors: RT-LAMP primers were designed based on nucleocapsid (N) protein gene sequence alignment of available at this time 46 sequences of TYRV isolates (12 sequences of the Polish isolates and 34 TYRV sequences retrieved from Gene Bank). All (15) the Polish isolates were detected by RT-LAMP. In presented manuscript we decided to show results of selected eight Polish isolates: TYRV-TK1, TYRV-TK2, TYRV-TK4, TYRV-BK1, TYRV-BK2, TYRV-H, TYRV-KOR1, TYRV-KOR2. To establish the sensitivity of the RT-LAMP assay one isolate (TYRV-H) was chosen.

  1. line 50: I would consider to cite https://doi.org/10.1371/journal.ppat.1002021 describing molecular mechanism behind symptom development in Y sat containing CMV infection.

Authors: The reference was added.

  1. Line 57: emphasize that TYRV has only been described from Poland, Iran and Kenya. It is not clear here and later on it was not straightforward to understand why you explain why the isolate from Kenya was not included in the analysis.

Authors: In our work, we performed bioinformatical analysis based on the full length sequence of nucleocapsid (N) protein. From 56 sequences of TYRV isolates deposited in GenBank only nucleotide sequence of Kenyan isolate is incomplete. Therefore, we decided to exclude this sequence from the set.

  1. Was RT-PCR done as a one step reaction? If you tested one sample for the presence of several viruses it would have been more valid to use the same cDNA for all of the virus test.

Authors: Yes, RT-PCRs were performed as a one-step reactions. Taking into account that we were looking for viruses with different genome organization using RT-PCR was more suitable than producing cDNAs. Using random hexamer primers to synthetize cDNAs is not always efficient. The RT-PCR reactions were valid in our laboratory and routinely used for the detection of different virus species.  

  1. line 144: instead of “deleted from the set” I would suggest “was not included in the analysis”

Authors: The sentence was changed.

  1. Table2: I would suggest to put it into the Supplementary materials

Authors: Table 2 was transferred to Supplementary materials.

  1. RT-LAMP: it seems that only one sets of LAMP primers were designed by the software.

It is not too much description on this primer design. Even if the software would have been designed the primers it could be aligned (not only to the Polish, but maybe for the Iranian and Kenya variants) to show how conservative the stretches of the sequence are what accommodate the primers. You made temperature gradient, but increased the temperature by quite high (3°C) steps. Wouldn’t it be better to slightly increase the temperature? keeping in mind that it is already a high temperature for disrupting the secondary structured RNAs? It is not stated how many times were the LAMP test carried out? This would be important to see the reproducibility of the assay. No characteristic parameters: sensitivity, reliability, reproducibility was investigated, this would be nice to do. You only used RNA for the LAMP test. As it can operate on crude extract it would further simplify its use. If you did not test that, please at least discuss this possibility.

Authors: RT-LAMP primers were designed based on nucleocapsid (N) protein gene sequence alignment of available at this time 46 sequences of TYRV isolates (12 sequences of the Polish isolates and 34 TYRV sequences retrieved from Gene Bank). The designed primers were complementary to genome region which is rather conservative among TYRV isolates The preliminary tests using designed primers revealed promising results therefore we performed further RT-LAMP optimalization. Based on our team experience with LAMP method and literature data usually the best results are obtained between 63-65°C. That’s why we tested temperature range as described in the manuscript. RT-LAMP products were obtained in all tested temperatures. Both, sensitivity as well reliability tests are described in the manuscript. The RT-LAMP test was carried our several times during optimalization and for the screening of newly delivered tomato samples.

Yes, only total RNAs were used for RT-LAMP test. The information about possibility of using crude extract to perform RT-LAMP was added to the discussion.

  1. Table 4: in this table I would change the order of the viruses and list them according to their incidence, as it was done in the discussion. This would make this table and the discussion parallel, and help to understand the numbers. I would include the satellites also, just after their helper virus.

Authors: The virus order in Table 4 was changed. In our opinion adding information about satRNAs (which were not detected in any cases) will make the table less readable.

  1. Figure 3 and results of the phylogenetic analysis. Your statement is that the strains cluster according to their geographical origin. The trees clearly show this, but I am curious: to which ones the isolate from Kenya would cluster. It would be possible to make a tree including 2-3 isolates from both 3 clusters (POL, IRN-1 and IRN-2) and cut them to the same size what the Kenya isolate has and make a tree.

Authors: Analysis including Kenyan isolate was carried out but not included in the manuscript. The TYRV-Loitoktok isolate from Kenya grouped within IRN-2 cluster as it was described in previous study of Golnaraghi et al. 2018.

  1. Comparison of RT-LAMP with RT_PCR. It is not clear from the shown pictures how the 10x increased sensitivity was determined. Where is nu9 on Fig 4/D?

Authors: Missing of 9th electrophoretic column on FIG 4D is our mistake of figure description. It was corrected. The description of sensitivity comparison is presented in method section.

  1. Negative result for the investigated viruses with RT-PCR can be the result of SNP variants which has mutation withing the primers sequence. This is something what you should at least discuss.

Authors: Additional information was added to the discussion.

  1. Is it possible to check back the seeds for the presence of the virus, to find out its origin? It is said in the M&M that the tomato cultivars were mainly Polish origin, while in the discussion the seeds imported from Israel and China are suspected as the source of the virus. This should be clarified.

Authors: We are sorry but this information regards tomato brown rugose fruit virus. It was added here by mistake, we deleted this sentence. There are no records of how TYRV was introduced to Iran and Kenya. However, it could have been introduced to Poland through infected tomato seedlings imported from Iran. We agree that screening seeds for virus presence is essential in plant protection. Seeds from abroad are tested by the Main Inspectorate of Plant Health and Seed Inspection in Poland and we did not perform additional studies including seeds from the Polish tomato cultivars. This type of studies are time and money consuming and we were not able to carry out them within this project. TYRV seed transmission requires further studies as there are not literature data regarding this subject.

Reviewer 2 Report

The authors of the manuscript entitled "Occurrence, Genetic Variability of Tomato Yellow Ring Ortho-2 Tospovirus Population and the Development of Reverse Tran-3 Scription Loop-Mediated Isothermal Amplification Assay for 4 Its Rapid Detection", during their systematic monitoring of the viruses infecting tomatoes, conducted for almost a decade, have identified occurrence of the new virus infecting tomatoes in Poland - TYRV. This critical finding prompted the authors to analyze the genetic structure of the Polish population of the virus compared to its original Iranian one. Interestingly, they have found that Polish isolates differentiate from the Iranian population. The last one divides into two distinct groups. Further analysis of the viral population revealed host impact on the virus evolution. However, except few clear cases (tomato vs. potato), the authors did not find significant differences between isolates. Finally, a detailed understanding of the virus variability at the nucleotide level and the potential direction evolutional force can shape it facilitated the authors' design of primers for the virus detection by isothermal RT-LAMP assay. This method is a sensitive, faster, and less effort-consuming alternative to the RT-PCR, which the author's results reflect well n the manuscript. I do not see any flaws and consider this paper well done, written, and essential. Therefore I recommend it publishing as it is.

Author Response

Reviewer 2

I do not see any flaws and consider this paper well done, written, and essential. Therefore I recommend it publishing as it is.

Authors: Thank you very much for your positive opinion!